# The Preliminary Analysis of Flavonoids in the Petals of *Rhododendron delavayi*, *Rhododendron agastum* and *Rhododendron irroratum* Infected with *Neopestalotiopsis clavispora*

**DOI:** 10.3390/ijms25179605

**Published:** 2024-09-04

**Authors:** Xiaoqian Shi, Yizhen Wang, Su Gong, Xianlun Liu, Ming Tang, Jing Tang, Wei Sun, Yin Yi, Jiyi Gong, Ximin Zhang

**Affiliations:** 1School of Life Sciences, Guizhou Normal University, Guiyang 550025, China; 2Key Laboratory of Environment Friendly Management on Alpine Rhododendron Diseases and Pests of Institutions of Higher Learning in Guizhou Province, Guizhou Normal University, Guiyang 550025, China; 3Key Laboratory of State Forestry Administration on Biodiversity Conservation in Karst Area of Southwest, Guizhou Normal University, Guiyang 550025, China; 4Key Laboratory of Plant Physiology and Development Regulation, Guizhou Normal University, Guiyang 550025, China

**Keywords:** alpine rhododendron, petal blight disease, flavonoids, naringenin, fungus

## Abstract

The petal blight disease of alpine Rhododendron severely impacts the ornamental and economic values of Rhododendron. Plant secondary metabolites play a crucial role in resisting pathogenic fungi, yet research on metabolites in alpine Rhododendron petals that confer resistance to pathogenic fungi is limited. In the present study, the secondary metabolites in *Rhododendron delavayi*, *R. agastum*, and *R. irroratum* petals with anti-pathogenic activity were screened through disease index analysis, metabolomic detection, the mycelial growth rate, and metabolite spraying experiments. Disease index analysis revealed that *R. delavayi* petals exhibited the strongest disease resistance, while *R. agastum* showed the weakest, both under natural and experimental conditions. UHPLC-QTOF-MS/MS analysis identified 355 and 274 putative metabolites in positive and negative ion modes, respectively. The further antifungal analysis of differentially accumulated baicalein, diosmetin, and naringenin showed their half-inhibitory concentrations (IC50) against *Neopestalotiopsis clavispora* to be 5000 mg/L, 5000 mg/L, and 1000 mg/L, respectively. Spraying exogenous baicalein, diosmetin, and naringenin significantly alleviated petal blight disease caused by *N. clavispora* infection in alpine Rhododendron petals, with the inhibition rates exceeding 64%. This study suggests that the screened baicalein, diosmetin, and naringenin, particularly naringenin, can be recommended as inhibitory agents for preventing and controlling petal blight disease in alpine Rhododendron.

## 1. Introduction

Alpine Rhododendron, evergreen shrubs or trees flourishing at altitudes above 1000 meters, are renowned for their large, vibrant, umbel-shaped flowers and glossy leaves [1,2]. The Baili Azalea Nature Reserve (BANR) in northwest Guizhou, China, boasts the largest known native Rhododendron forest in the country, hosting around 40 species of alpine Rhododendron [3,4]. Among these, *Rhododendron delavayi*, *R. agastum*, and *R. irroratum* are dominant, each distinguished by unique floral characteristics [3,5]. *R. delavayi* showcases deep red, bell-shaped flowers and serves as a crucial breeding parent in the Rhododendron genus. *R. irroratum*, which is abundant in resources, features pale yellow or white petals adorned with yellow-green to light purple-red spots. *R. agastum*, which is believed to be the hybrid offspring of *R. delavayi* and *R. irroratum* [6,7], exhibits a bell-shaped funnel corolla with pink petals and purple-red spots inside. The striking colors and captivating flower shapes of these three species have garnered global attention from horticulturists.

In recent years, the BANR has experienced a surge in petal blight disease across numerous species, triggered by pathogenic infestations [8]. This disease manifests initially as small brown lesions on petals, which progressively expand and darken, ultimately leading to petal wilting and decline [9]. The detrimental effects extend beyond mere aesthetics, severely impacting plant reproduction. The proliferation of petal blight disease has inflicted substantial economic losses on the region’s ornamental flower-based tourism industry, while also diminishing biodiversity within the ecosystem. Consequently, investigating the disease resistance mechanisms in *Rhododendron* is paramount for developing resilient cultivars and safeguarding this valuable natural resource.

Plants have evolved intricate defense mechanisms to counteract various biological stressors, including fungal pathogens [10,11]. These defenses encompass the production of secondary metabolites possessing antifungal properties, capable of limiting pathogen proliferation [12,13,14,15]. Flavonoids, a key class of antimicrobial compound, have been documented for their defensive role against fungi. For instance, farrerol, a flavonoid detected in *R. hainanense*, can inhibit the growth of pathogenic fungi (*Fusarium oxysporum f.* sp. *niveum, Colletotrichum gloeosporioides, Penicillium italicum, Rhizoctonia solani, Fusarium oxysporum f.* sp. *Cubenserace,* and *Phytophthora melonis*) [16]. Quercetin, kaempferol, and their derivatives, the flavonoid extracts from Ginkgo biloba leaves, have inhibitory effects on *P. expansum* [17]. α, β-dihydroxanthohumol and 8-prenylnaringenin from the spent hops of *Humulus lupulus* have inhibitory effects on *Botrytis cinerea* and *F. oxysporum* [18]. The extracts (quercetin, naringenin, and kaempferol) from *Acacia saligna* petals inhibit the growth of *F. culmorum*, *R. solani*, and *P. chrysogenum* [19]. Additionally, the accumulation of chlorogenic acid, apigenin, and quercetin in infected *R. agastum* petals suggests their involvement in combating petal blight disease [20]. Despite these findings, research specifically investigating the antifungal capabilities of flavonoids in alpine Rhododendron petals against pathogen infestation remains scarce.

This study employed a multi-faceted approach, utilizing disease index analysis, metabolomics detection, mycelial growth rate analysis, and secondary metabolite spraying experiments on the petals of *R. delavayi*, *R. agastum*, and *R. irroratum* (Figure 1) to test the following hypotheses: (1) *R. delavayi* petals exhibit superior disease resistance compared to *R. agastum* and *R. irroratum*; (2) *R. delavayi* petals possess a higher content of flavonoid metabolites; (3) certain flavonoid metabolites present in greater abundance demonstrate antifungal properties. Through this comprehensive investigation, the aim is to identify potential plant secondary metabolites, thereby providing valuable leads for the future development of effective strategies to control petal blight disease.

## 2. Results

### 2.1. Analysis of Disease Severity in Three Alpine Rhododendron Petals

In this study, disease incidence and the disease index were employed to assess petal blight disease severity in three alpine Rhododendron species within the BANR. The results revealed the lowest disease incidence and disease index (15.7% and 22.53) in *R. delavayi* petals and the highest (88.2% and 70.79) in *R. agastum* (Figure 2A,C). Similarly, the disease index of *R. delavayi* petals (22.53) was significantly lower than that of *R. irroratum* (42.92) (Figure 2C).

Additionally, pathogen infection experiments were conducted on the petals of the three species under laboratory conditions. These experiments demonstrated that the *R. delavayi* petals exhibited significantly lower disease incidence and percent disease index (43.8% and 45.40) compared to those of *R. agastum* (76.3%, 56.64) and *R. irroratum* (62.2%, 52.25) (Figure 2B,D). Collectively, these findings indicate that *R. delavayi* petals possess strong disease resistance, while *R. agastum* petals are the most susceptible to petal blight disease.

### 2.2. Detection and Identification of Metabolites in the Petals of Three Alpine Rhododendrons

The petals from the three species were analyzed using UHPLC-QTOF-MS/MS. The resulting chromatograms of 18 petal samples displayed good reproducibility, confirming the stability and reliability of the operating conditions (Appendix A). In the quality control samples, the relative deviations of the L-2-chlorophenylalanine internal standard were 6.66% and 2.37% in the positive and negative ion modes, respectively, underscoring the system’s high stability. A total of 1732 and 1994 peaks were extracted in the positive and negative ion modes, respectively, culminating in the identification of 355 and 274 putative metabolites (Appendix A). These metabolites were categorized into primary and secondary metabolites. The secondary metabolites were further classified into eight groups, including 20 alkaloids, 13 coumarins, 49 flavonoids, 15 lignans, 17 phenylpropanoids, 20 terpenoids, and others.

### 2.3. Principal Component Analysis (PCA) and Orthogonal Projections to Latent Structure-Discriminant Analysis (OPLS-DA) of Metabolites in Three Alpine Rhododendron Petals

Utilizing UHPLC-QTOF-MS/MS analysis, distinct metabolite profiles were observed among the three Rhododendron species. PCA illustrated clear separation in both the positive and negative ion modes. In the positive ion mode, PC1 and PC2 explained 44.3% and 14.6% of the total variance, respectively (Figure 3A), while in the negative ion mode, these components accounted for 45.2% and 16.2% (Figure 3C). OPLS-DA further validated these distinctions, with the first two components explaining 41.4% and 16.9% of the variance in the positive ion mode (Figure 3B) and 41.5% and 19.7% in the negative ion mode (Figure 3D), respectively. These results confirm significant metabolite differentiation among the three alpine Rhododendron petal types.

### 2.4. Differentially Abundant Metabolite Analysis

Differentially abundant metabolite analysis employing the thresholds of *p* < 0.05 and VIP > 1 unveiled unique metabolite signatures for each species (Figure 4). In the positive ion mode, *R. delavayi* exhibited 259 upregulated and 702 downregulated different metabolites compared to those of *R. irroratum* (Figure 4A), while *R. delavayi* showed 322 upregulated and 452 downregulated different metabolites compared to those of *R. agastum* (Figure 4B). Notably, *R. agastum* demonstrated 175 upregulated and 671 downregulated different metabolites compared to those of *R. irroratum* (Figure 4C). In the negative ion mode, consistent trends were observed, with *R. delavayi* displaying 405 upregulated and 746 downregulated different metabolites compared to those of *R. irroratum* (Figure 4D), *R. delavayi* exhibiting 435 upregulated and 507 downregulated different metabolites compared to those of *R. agastum* (Figure 4E), and *R. agastum* showcasing 283 upregulated and 714 downregulated different metabolites compared to those of *R. irroratum* (Figure 4F).

To examine the differentially abundant metabolite patterns across various metabolite classes, the screened metabolites were classified, and the proportion of each class within the total identified metabolites was determined. In the upregulated metabolites, compared to *R. irroratum*, flavonoids accounted for the highest proportion at 42.86%, followed by the coumarins (38.46%), simple phenylpropanoids (29.41%), lignans (26.67%), alkaloids (13.33%), and terpenes (10.00%) in *R. delavayi* (Figure 5A); compared to *R. irroratum*, in the upregulated metabolites, simple phenylpropanoids accounted for 17.65%, followed by the alkaloids (16.67%), lignans (13.33%), flavonoids (12.24%), and coumarins (7.69%) in *R. agastum* (Figure 5B). Compared to *R. agastum*, in the upregulated metabolites, flavonoids accounted for 51.2%, followed by the simple phenylpropanoids (41.67%), simple phenylpropanoids (41.67%), lignans (40.00%), coumarins (30.77%), alkaloids (13.33%), and terpenes (10.00%) in *R. delavayi* (Figure 5C). These results indicate that flavonoids in *R. delavayi* have the highest proportion in the upregulated metabolites.

### 2.5. Screening of Flavonoid Metabolites with Potential Antifungal Effects

Integrating the observed disease resistance of *R. delavayi* petals with the differentially abundant metabolite accumulation patterns, the relative quantitative analysis of flavonoid metabolites was performed to identify the potential antifungal agents. In the positive ion mode, the *R. delavayi* petals exhibited higher relative contents of procyanidin A1, procyanidin B2, procyanidin A2, baicalein, hesperetin, hyperoside, and diosmetin compared to those of *R. agastum* and *R. irroratum* (Figure 6 and Appendix A). Additionally, in the negative ion mode, the *R. delavayi* petals contained the highest relative amounts of apigenin, naringenin, diosmetin, quercetin, taxifolin, isoquercetin, kaempferol, procyanidin A, and procyanidin B, surpassing those of both *R. agastum* and *R. irroratum* (Figure 7 and Appendix A). A literature review confirmed that several of these secondary metabolites, namely baicalein, diosmetin, apigenin, naringenin, quercetin, taxifolin, and kaempferol, have documented antifungal properties (Table 1). These findings highlight these flavonoids as prime candidates for further investigation into their potential roles in conferring disease resistance to *R. delavayi* petals.

### 2.6. Inhibitory Concentrations of Exogenous Metabolites on the Mycelial Growth of N. clavispora

Building upon our previous finding that exogenous apigenin effectively inhibits pathogen infection in *R. simsii* petals, we extended our antifungal analysis to baicalein, diosmetin, naringenin, quercetin, taxifolin, and kaempferol. Taxifolin and kaempferol did not significantly inhibit the mycelial growth of *N. clavispora,* whereas baicalein, diosmetin, naringenin, and quercetin demonstrated significant inhibitory effects (Appendix A).

Subsequently, we determined the half-inhibitory concentrations (IC50) of baicalein, diosmetin, naringenin, and quercetin, baicalein, and diosmetin at concentrations ranging from 750 to 5000 mg/L effectively suppress mycelial growth (Figure 8A,C). At 5000 mg/L, the mycelial growth inhibition rates of baicalein and diosmetin reached 52.19% and 55.86% after 3 days and 54.56% and 48.78% after 7 days, respectively (Figure 8B,D). Naringenin, at concentrations of 250–3000 mg/L, also inhibited mycelial growth, achieving inhibition rates of 55.35% and 60.86% after 3 and 7 days, respectively, at a concentration of 1000 mg/L (Figure 8E,F). In contrast, quercetin, while showing an inhibitory effect at 750–3000 mg/L, did not exceed a 20% inhibition rate (Appendix A). These results establish the IC50 values of baicalein, diosmetin, and naringenin for *N. clavispora* mycelial growth as 5000 mg/L, 5000 mg/L, and 1000 mg/L, respectively.

### 2.7. Inhibitory Effects of Baicalein, Diosmetin, and Naringenin on Petal Blight in Alpine Rhododendron

To further investigate the potential of baicalein, diosmetin, and naringenin in mitigating petal blight caused by *N. clavispora* infection in alpine Rhododendron, experiments were conducted on the petals of *R. delavayi*, *R. agastum*, and *R. irroratum*. Four days post-inoculation with *N. clavispora*, the lesion areas were measured as 25.29 mm^2^, 34.13 mm^2^, and 39.64 mm^2^ for the *R. delavayi*, *R. agastum*, and *R. irroratum* petals, respectively (Figure 9A). Upon spraying with 5000 mg/L baicalein, the lesion areas were significantly reduced to 7.02 mm^2^, 8.50 mm^2^, and 9.87 mm^2^ in the *R. delavayi*, *R. agastum*, and *R. irroratum* petals, respectively (Figure 9B). Similarly, spraying with 5000 mg/L diosmetin resulted in lesion areas of 6.23 mm^2^, 11.65 mm^2^, and 6.44 mm^2^ for the three species, respectively (Figure 9B). Moreover, the application of 1000 mg/L naringenin led to significant reductions in the lesion areas, measuring 8.59 mm^2^, 12.16 mm^2^, and 8.79 mm^2^ in the *R. delavayi*, *R. agastum*, and *R. irroratum* petals, respectively (Figure 9B). Compared to the control, spraying with baicalein, diosmetin, and naringenin achieved disease inhibition rates exceeding 64% in all the three species (Appendix A). These results unequivocally demonstrate the efficacy of exogenous baicalein, diosmetin, and naringenin in substantially reducing petal blight caused by *N. clavispora* infection in the alpine Rhododendron petals.

## 3. Discussion

### 3.1. Evaluation of Disease Resistance in Wild Trees and Flowers Can Be Conducted by Counting the Number of Diseased Flowers

The disease index serves as an indicator of a plant’s resilience to pathogen infection, and its use has been documented in various plant species, such as tomato, *Anthurium andraeanum*, and rubber tree [28,29,30]. While the previous studies often relied on lesion area calculation to determine the disease index [28,29,31,32], this approach proved challenging for the tree species investigated in our research due to the difficulty in accurately measuring the petal lesion areas. Therefore, we opted to utilize the relationship between infected flowers and total flowers to calculate the disease index. The analysis of the three species in their natural habitat revealed that *R. delavayi* exhibited the lowest petal infection rate and disease index, while *R. agastum* displayed the highest infection rate (Figure 2A,C), signifying stronger disease resistance in *R. delavayi*.

Furthermore, laboratory infection experiments with *N. clavispora* were conducted to estimate the lesion areas and assess disease resistance (Figure 2). The results demonstrated a strong correlation between the number of infected flowers and the lesion area in evaluating disease resistance, corroborating the finding that *R. delavayi* possesses superior disease resistance among the three species.

### 3.2. Significant Differences in Endogenous Flavonoids Exist among Three Alpine Rhododendrons

The previous studies have established *R. agastum* as the hybrid offspring of *R. delavayi* and *R. irroratum* [6]. This hybrid is easily distinguishable from its parent species by the petal color (Figure 1). While significant differences in the secondary metabolites have been reported between hybrid offspring and their parent species in some cases (e.g., tea plant maternal Tieguanyin and paternal Huangdan, *Jacobaea vulgaris* and *J. aquatica*; and *Betula pendula* and *Betula pubescens hybrids*) [33,34,35], it remains unclear whether such differences exist in the petals of *R. agastum* and its parents.

Our analysis of petal disease resistance among the three alpine Rhododendrons, coupled with metabolite detection and analysis, revealed a separation trend in the PCA analysis of the metabolites (Figure 3), indicating substantial metabolic differences despite their genetic relatedness. Differentially abundant metabolite analysis highlighted the significant accumulation of flavonoid secondary metabolites in *R. delavayi* compared to those of *R. agastum* and *R. irroratum* under both the detection modes (Figure 6 and Figure 7). Given the superior disease resistance of the *R. delavayi* petals (Figure 2), this suggests that flavonoid accumulation may contribute to defense against pathogenic infection [25,36]. The relative quantification of flavonoid metabolites identified 21 differentially accumulated flavonoids in *R. delavayi*. Literature review confirmed the antifungal properties of flavones such as baicalein can inhibit against *C. albicans*, *C. tropicalis* and *C. parapsilosis*, *C. glabrata*, diosmetin against *Bacillus subtilis* and *Trichophyton rubrum* [22,37]; apigenin against *Alternaria tenuissima* and *C. albicans* [23], naringenin against *Phytophthora nicotianae*, *P. citrophthora* and *P. palmivora* [38,39]; quercetin against *C. parapsilosis* and *Colletotrichum gloeosporioides* [24,26], taxifolin against *Mycosphaerella graminicola*, *Diplodia pinea* [27,40]; and kaempferol against *C. parapsilosis* and *Botrytis cinerea* [24,41]. Inhibition analysis using *N. clavispora*, a fungus isolated from *R. delavayi* petals, revealed no inhibitory effects of taxifolin and kaempferol. It is important to note that antifungal agents can exhibit broad-spectrum or specific activities [42]. For example, kaempferol inhibits the leaf blight pathogen *Dothiorella gregaria* in Ginkgo biloba, but not two other pathogens (*Alternaria tenuissima* and *Botryosphaeria dothidea*) [43]. Therefore, further research is needed to ascertain whether taxifolin and kaempferol possess antifungal effects against other fungi.

### 3.3. Antifungal Effects of Baicalein, Diosmetin, and Naringenin on N. clavispora

The half-inhibitory concentrations (IC50) of baicalein, diosmetin, and naringenin were determined to be 5000 mg/L, 5000 mg/L, and 1000 mg/L, respectively (Figure 7). While all the three metabolites exhibited inhibitory effects on *N. clavispora* growth, the significantly lower IC50 of naringenin suggests its superior efficacy in suppressing this pathogenic fungus. This is different from the inhibitory effect of these metabolites on *Curvularia lunata* [44]; this observation aligns with the concept of metabolite specificity in fungal inhibition [42]. For instance, baicalein demonstrates greater inhibitory activity against *Acidovorax avenae* subsp. *cattlyae, Clavibacter michiganensis* subsp. *michiganensis*, and *Ralstonia solanacearum* [45]; diosmetin effectively inhibits *B. subtilis* and *T. rubrum* [23], while naringenin suppresses the growth of *F. solani*, *M. canis*, *T. logifusus*, and *C. glabrata* [46]. Although quercetin, at varying concentrations, also demonstrated an inhibitory effect on *N. clavispora* mycelial growth, its inhibition rates remained below 20%, even at 3000 mg/L (Appendix A). Consequently, quercetin may not be a suitable candidate for developing control agents against this particular fungus.

Following the inoculation of *N. clavispora* on the petals of the three Rhododendron species, spraying with 5000 mg/L baicalein, 5000 mg/L diosmetin, or 1000 mg/L naringenin resulted in inhibition rates exceeding 64% compared to those of the control treatment (Figure 8 and Appendix A). Considering the cost-effectiveness of metabolites, naringenin emerges as the preferred control agent for petal blight caused by *N. clavispora* in alpine Rhododendron. In the current study, baicalein, diosmetin, and naringenin were screened and tested for their ability to inhibit *N. clavispora* pathogenicity. Previous literature reported that baicalein inhibits glycolysis by targeting protein Eno1 and biofilm formation to restrict fungal growth [47,48]. In addition, baicalein can also induce increased reactive oxygen species (ROS) levels, thereby inducing pathogen apoptosis [49]. Moreover, naringenin can induce the outbreak of ROS and the accumulation of salicylic acid, activate the expression of resistance genes, and thus exhibit resistance to pathogens [38]. However, current research has not explored the possible mechanisms. Further research is warranted to unravel how these metabolites interfere with fungal growth and development, thereby contributing to the development of more targeted and effective disease management strategies. In addition, in the experiment of spraying metabolites after pathogen infection on petals, the lesion area of the control treated *R. irroratum* petals was higher than that of the *R. agastum* (Figure 9), which is slightly inconsistent with our natural conditions (Figure 2). We found that 0.01 mol/L NaOH solution had a slight effect on the formation of lesions on petals, especially on the white petals, but this effect could not be distinguished from the lesions, resulting in a slight increase the lesion area of the *R. irroratum* petals sprayed with 0.01 mol/L NaOH.

## 4. Materials and Methods

### 4.1. Plant Materials and N. clavispora

In April 2023, petals from *R. delavayi*, *R. irroratum*, and *R. agastum* were collected from the BANR in Guizhou Province, China (27°23′ N, 106°86′ E) (Figure 1). For each replicate sample, 5–6 fresh, healthy petals were selected from different branches of individual trees. A total of 6 replicate samples, each from a distinct tree, were collected. The petals were promptly wrapped in aluminum foil, immersed in liquid nitrogen, and stored at −80 °C for subsequent analysis.

To definitively identify the pathogenic fungus responsible for petal blight disease in the Rhododendron genus, a comprehensive series of studies was undertaken, encompassing the isolation, inoculation, microscopic observation, and molecular biological identification of the infected petals. These efforts culminated in the successful isolation and molecular identification (Genomic DNA extraction, ITS PCR amplification and sequencing) of a pathogenic strain (MR-001) as *N*. *clavispora*. Pathogenicity tests confirmed this fungus’s ability to infect petals of the *Rhododendron* species, inducing blight disease symptoms. The *N. clavispora* strain is preserved in the reservoir (Haier Group Co., Ltd. Qingdao, China) in Key Laboratory of Plant Physiology Development and Regulation in Guizhou Province.

### 4.2. Cultivation of Pathogenic Fungi

The fungus activation process involved using a sterile 5 mm diameter drill bit (sterilized at 121 °C for 30 min) to extract fungal blocks from the edge of the colony, ensuring the hyphae faced one side of the culture medium. These blocks were then incubated at 28 °C for 7 days. Following two generations of activation, a fungal mycelial suspension of *N. clavispora* was prepared by stirring and diluting the mycelium with sterile water to a final concentration of 0.025 mg/mL.

### 4.3. Fungal Infection and Disease Analysis

A sterile scalpel with a 2 mm diameter was used to puncture the petals, and 7 μL of fungal suspension was introduced into the wound, with sterile water serving as the control. Blooming *R. delavayi*, *R. agastum*, and *R. irroratum* were selected, and their petals were infected with the pathogen using established laboratory infection methods. The inoculated petals were placed in a culture room maintained at 24–27 °C and 90% relative humidity.

For *R. delavayi*, *R. agastum*, and *R. irroratum* grown under natural conditions, disease incidence was calculated as (number of diseased flowers/total number of flowers) × 100%. Disease assessment followed a 0–6 scale: 0 (asymptomatic), 1 (1–2% infection), 2 (3–5%), 3 (6–29%), 4 (30–39%), 5 (40–49%), and 6 (≥50%) [50]. The disease index was then calculated using the formula: disease index = Σ (number of plants at each level × relative level value)/(total number of plants surveyed × highest level value) × 100 [28].

For fungal infection under laboratory control conditions, photographs of the treated petals were taken 4 days post-inoculation, and the lesion areas were calculated using ImageJ software (1.8.0). Disease resistance among the species was evaluated by comparing the lesion areas.

### 4.4. Determination of the Inhibitory Rate of Flavonoids on the Mycelial Growth of N. clavispora

Baicalein (Cat: SB8010), diosmetin (Cat: SD8390), naringenin (Cat: SN8020), quercetin (Cat: SQ8030), taxifolin (Cat: ST8050), and kaempferol (Cat: SK8030) were purchased from Beijing Solarbio Technology Co., Ltd., Beijing, China. They were dissolved in 10 mL of 1 mol/L NaOH solution after being subjected to 30 min of ultraviolet irradiation, adjusting the concentrations based on their content in the PDA medium. Concurrently, 37 g of PDA powder was dissolved in 990 mL of distilled water and sterilized at 121 °C for 30 min. After cooling to approximately 50 °C, the metabolite solutions were sterile-filtered (0.22 μm) and added to the PDA solution, creating PDA media with varying concentrations. A control medium was prepared using 10 mL of 1 mol/L NaOH solution in PDA [51].

Activated mycelial discs (5 mm diameter) were punched from the colony edge, with the mycelial side facing the medium, and placed onto PDA media containing different compound concentrations. The cultures were incubated at 28 °C, and colony diameters were measured on days 3, 5, and 7 using the cross method, with mycelial disc areas calculated using ImageJ software (1.8.0). The inhibition rate of mycelial growth (%) was calculated as follows: [(control colony diameter − mycelial disc diameter) − (treatment colony diameter − mycelial disc diameter)]/(control colony diameter − mycelial disc diameter) × 100 [51].

### 4.5. Inhibition of Petal Blight by Metabolites in Rhododendron Species

Following the established petal infection method, after 24 h of exposure to the *N. clavispora* mycelial suspension, the infected petals were sprayed daily with either a control solution (0.01 mol/L NaOH) or metabolite solutions dissolved in 0.01 mol/L NaOH. The treated petals were photographed, and the lesion areas were quantified using ImageJ software. The inhibition rate was calculated as follows: inhibition rate (%) = (average lesion area of control − average lesion area of treatment)/average lesion area of control × 100 [51].

### 4.6. Detection and Identification of Metabolites in the Petals of Alpine Rhododendrons

Metabolites were extracted following the method of Dunn et al. [52]. Briefly, 50 mg of petal samples was ground and transferred to EP tubes. Subsequently, 1000 μL of extraction solvent containing internal standards (V_methanol:acetonitrile:water_ = 2:2:1, with 2 μL mL^−1^ internal standard) was added. The samples were sonicated (5 min) and centrifuged (4 °C, 12,000 rpm, 15 min). Then, 300 μL of reconstitution solvent (V_acetonitrile:water_ = 1:1) was added, followed by another round of sonication (10 min) and centrifugation (4 °C, 12,000 rpm, 15 min). Finally, 75 μL of the supernatant was collected for analysis.

Metabolite analysis was conducted using a UHPLC system (1290, Agilent Technologies, Santa Clara, CA, USA) coupled with a Triple TOF 6600 (Q-TOF, AB Sciex, Framingham, MA, USA) mass spectrometer. Metabolite separation was achieved on a UPLC BEH Amide column (1.7 μm, 2.1 × 100 mm, Waters) using a mobile phase composed of 25 mM NH4Ac and 25 mM NH4OH aqueous solution (pH 9.75) (A) and acetonitrile (B). The elution gradient was as follows, 0–0.5 min, 95% B; 7 min, 65% B; 8 min, 40% B; 9 min, 40% B; and 9.1–12 min, 95% B, with a flow rate of 0.5 mL min^−1^ and an injection volume of 1 μL.

For mass spectrometry analysis, a Triple TOF 6600 mass spectrometer, in conjunction with Analyst TF 1.7 software (AB Sciex), was employed to acquire the primary and secondary mass spectrometry data. In each acquisition cycle, molecular ions with intensities exceeding 100 were selected for corresponding secondary mass spectrometry data acquisition under the following conditions: collision energy, 30 eV; nebulizer gas pressure (GS1), 60 Psi; auxiliary gas pressure, 60 Psi; curtain gas pressure, 35 Psi; temperature, 600 °C; and spray voltage, 5000 V (positive ion mode) or −4000 V (negative ion mode).

### 4.7. Data Preprocessing and Annotation

Peak extraction, baseline correction, deconvolution, peak integration, and peak alignment were performed on the mass spectrometry data using ChromaTOF software (V4.3x, LECO, Bellefonte, PA, USA). Substance identification utilized the LECO-Fiehn Rtx5 database, employing mass spectrometry matching and retention time index matching. The peaks with a detection rate below 50% or RSD > 30% in the QC samples were excluded.

For the UHPLC-QTOF-MS/MS data, the raw mass spectrometry data were converted to mzXML format using ProteoWizard software (3.0.9134). XCMS was then employed for retention time correction, peak identification, peak extraction, peak integration, and peak alignment, with minfrac set to 0.5 and the cutoff set to 0.6. Peak identification was accomplished using self-written R programs and a custom-built secondary mass spectrometry database.

### 4.8. Principal Component Analysis (PCA) and Orthogonal Projections to Latent Structure-Discriminant Analysis (OPLS-DA)

The preprocessed data were utilized to generate a three-dimensional data matrix in CSV format, comprising the metabolite names, sample information (6 biological replicates per petal sample), and raw abundances (peak areas of each identified metabolite). These matrix data were then subjected to principal component analysis (PCA) and orthogonal projections to latent structure-discriminant analysis (OPLS-DA) using MetaboAnalyst 4.0 online software, incorporating three types of normalization, log transformation, and auto-scaling.

### 4.9. Differentially Abundant Metabolite Screening

The differentially abundant metabolites among the three species were selected based on a Student’s *t*-test with a *p*-value < 0.05 and a Variable Importance in the Projection (VIP) score > 1.

### 4.10. Statistical Analysis

The data are expressed as mean ± standard deviation (SD) and were analyzed used the LSD test with one-way analysis of variance (ANOVA). Analysis was carried out in at least three replicates for each sample, and a *p* < 0.05 was considered statistically significant.

## 5. Conclusions

In the present study, UHPLC-QTOF-MS/MS metabolomics technology was employed to compare the petals of disease-resistant *R. delavayi* with those of more susceptible *R. agastum* and *R. irroratum*. The results revealed the significant enrichment of flavonoids in the *R. delavayi* petals, suggesting a potential correlation with its enhanced disease resistance. Further antifungal analysis demonstrated that baicalein, diosmetin, and naringenin exerted substantial inhibitory effects on *N. clavispora*, with naringenin exhibiting superior inhibition compared to that of the other two compounds. Moreover, the exogenous application of baicalein, diosmetin, and naringenin significantly alleviated petal blight caused by *N. clavispora* infection in all the three species. These findings underscore the potential of baicalein, diosmetin, and naringenin, particularly naringenin, as promising biological control agents for fungal diseases in alpine Rhododendron. However, the precise inhibitory mechanisms underlying these compounds’ actions remain to be elucidated, warranting further investigation into their specific modes of action against fungal pathogens.

## Figures and Tables

**Figure 1 ijms-25-09605-f001:**
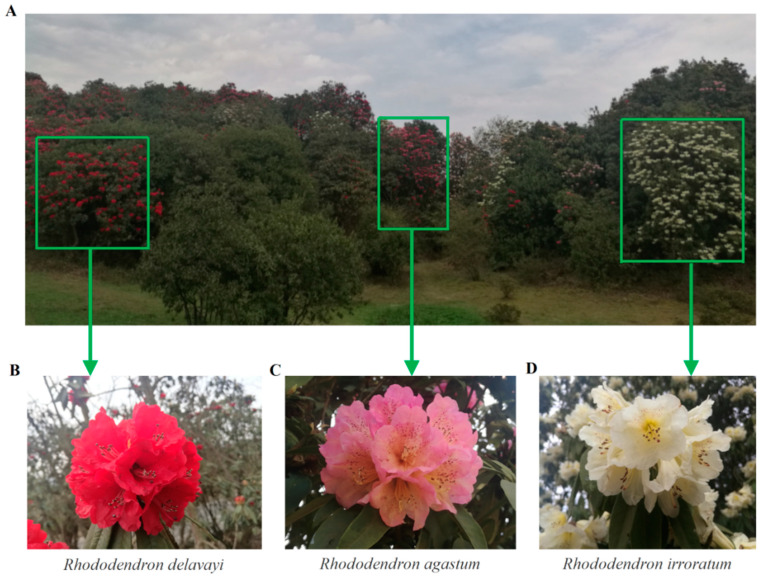
The sampling of flowers from *R. delavayi*, *R. agastum*, *R. delavayi*, and *R. irroratum*. (**A**) The three Rhododendron populations were located in the natural habitats of the Baili Azalea Nature Reserve. (**B**) *R. delavayi* has carmine-colored flowers. (**C**) *R. agastum* has pink flowers. (**D**) *R. irroratum* has jasmine-colored flowers.

**Figure 2 ijms-25-09605-f002:**
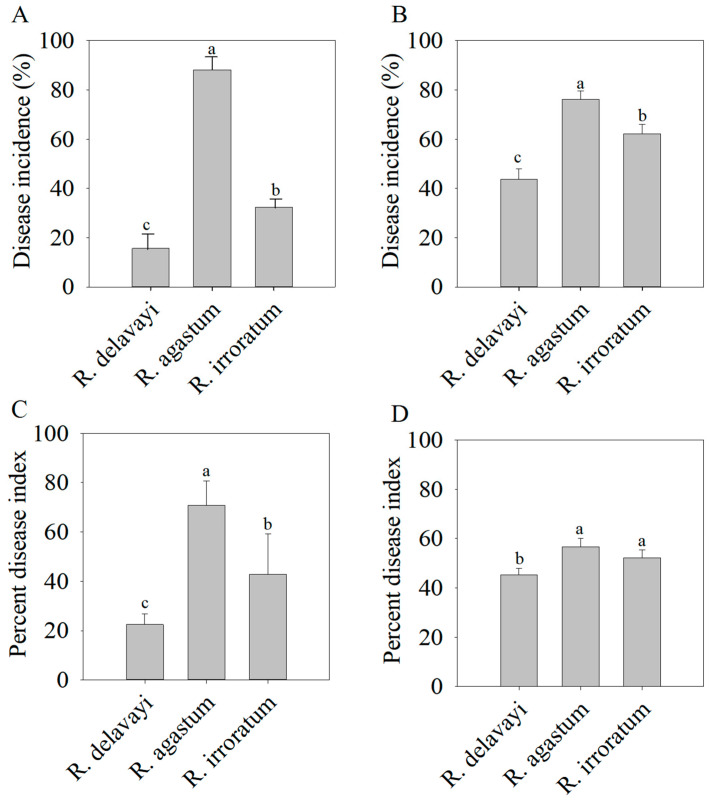
The disease incidence and percent disease index of *R. delavayi*, *R. agastum*, and *R. irroratum*. (**A**,**C**) illustrate the disease incidence and percentage disease index of the petals in the three species under natural conditions, respectively. (**B**,**D**) represent the disease incidence and percentage disease index of the petals of the three species under experimental conditions. Significance was analyzed using one-way analysis of variance (ANOVA) among the three species and is indicated by the different lowercase letters (*p* < 0.05).

**Figure 3 ijms-25-09605-f003:**
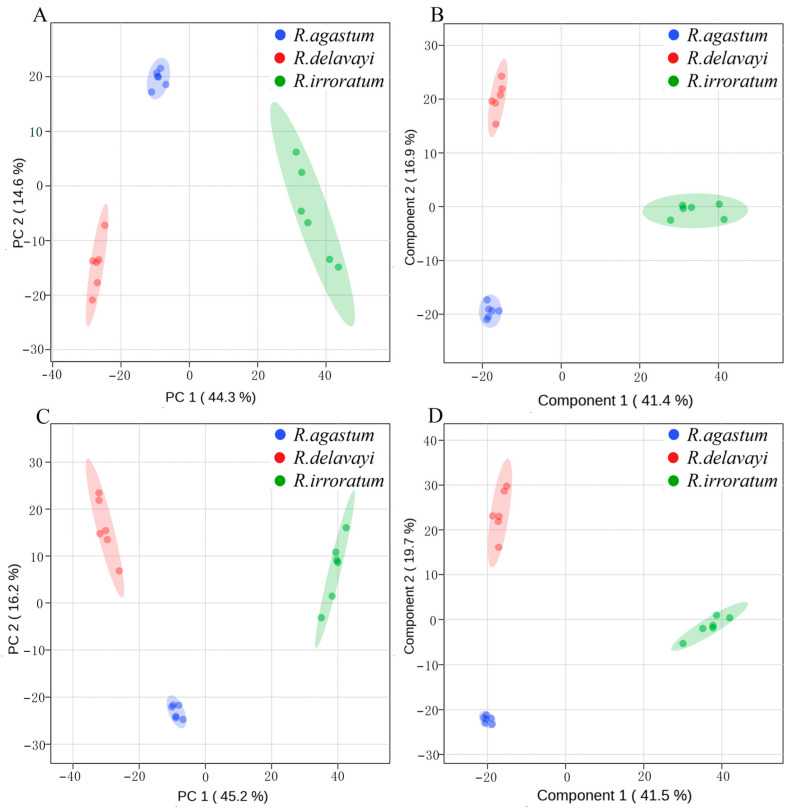
The PCA and OPLS-DA scores for the petal metabolites of *R. delavayi*, *R. agastum*, and *R. irroratum* detected using the UHPLC-QTOF-MS/MS platform. (**A**) shows the positive ion mode in PCA. (**C**) shows the negative ion mode in PCA. (**B**) shows the positive ion mode in OPLS-DA. (**D**) shows the negative ion mode in OPLS-DA. The red, blue, and black circles indicate *R. delavayi*, *R. agastum*, and *R. irroratum*, respectively, within the 95% confidence interval.

**Figure 4 ijms-25-09605-f004:**
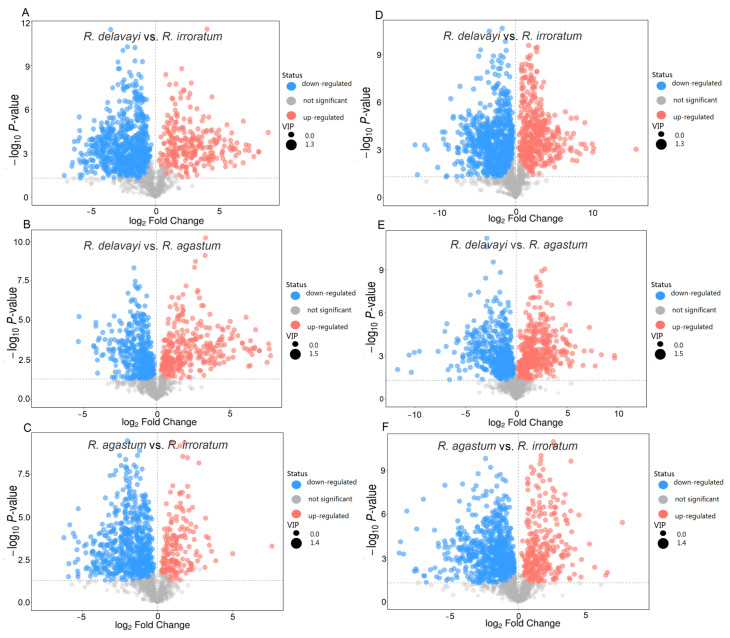
The volcano plots of differentially accumulated metabolites in the three species using UHPLC-QTOF-MS. (**A**–**C**) shows the positive ion mode. (**D**–**F**) shows the negative ion mode. The x-axis represents the mean of the log_2_ fold-change (FC) value, while the y-axis depicts the negative logarithm of the *p*-values. Each circle on the plot represents an individual metabolite. The red and green circles indicate the statistically significant changes in the metabolites, with the red circles representing the upregulated metabolites, the green circles representing the downregulated metabolites, and the gray circles indicating no significant changes in the metabolite levels.

**Figure 5 ijms-25-09605-f005:**
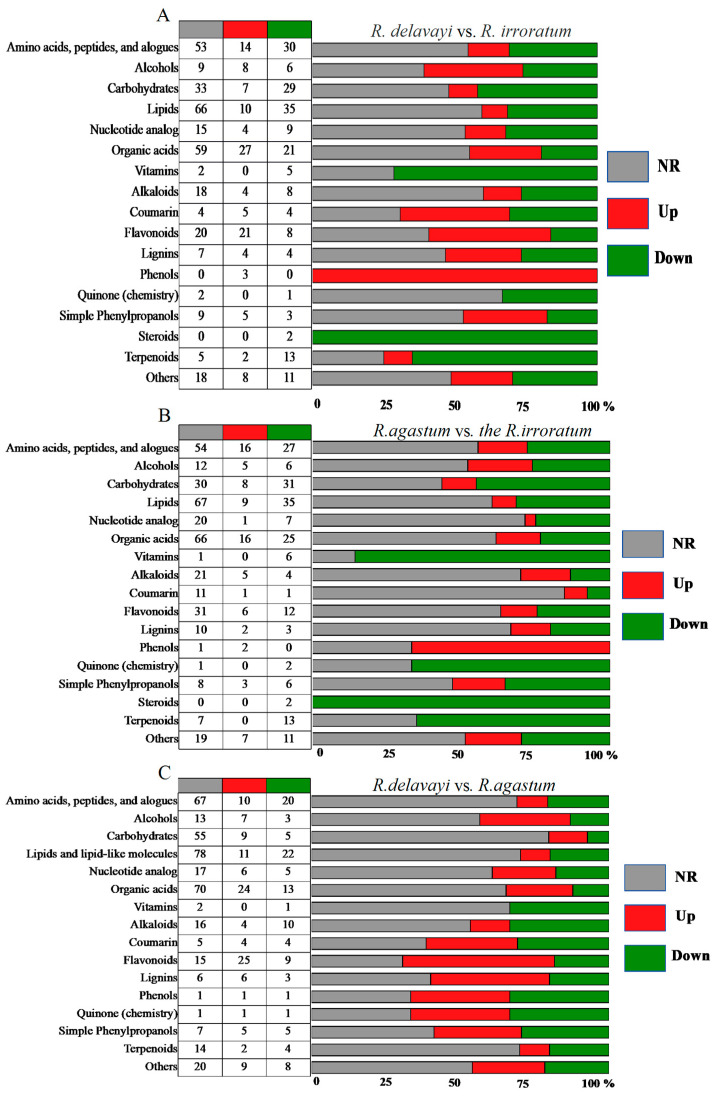
The UHPLC-QTOF-MS/MS platform was used to analyze the differential metabolites of *R. delavayi*, *R. agastum*, and *R. irroratum*. (**A**) represents the differentially accumulated metabolites of *R*. *delavayi* vs. *R. irroratum*; (**B**) represents the differentially accumulated metabolites of *R. agastum* vs. *R. irroratum*; (**C**) represents the differentially accumulated metabolites of *R. delavayi* vs. *R. agastum*. The data show the number and percentage of upregulated (red), downregulated (green), and non-responsive (gray) metabolites in each category.

**Figure 6 ijms-25-09605-f006:**
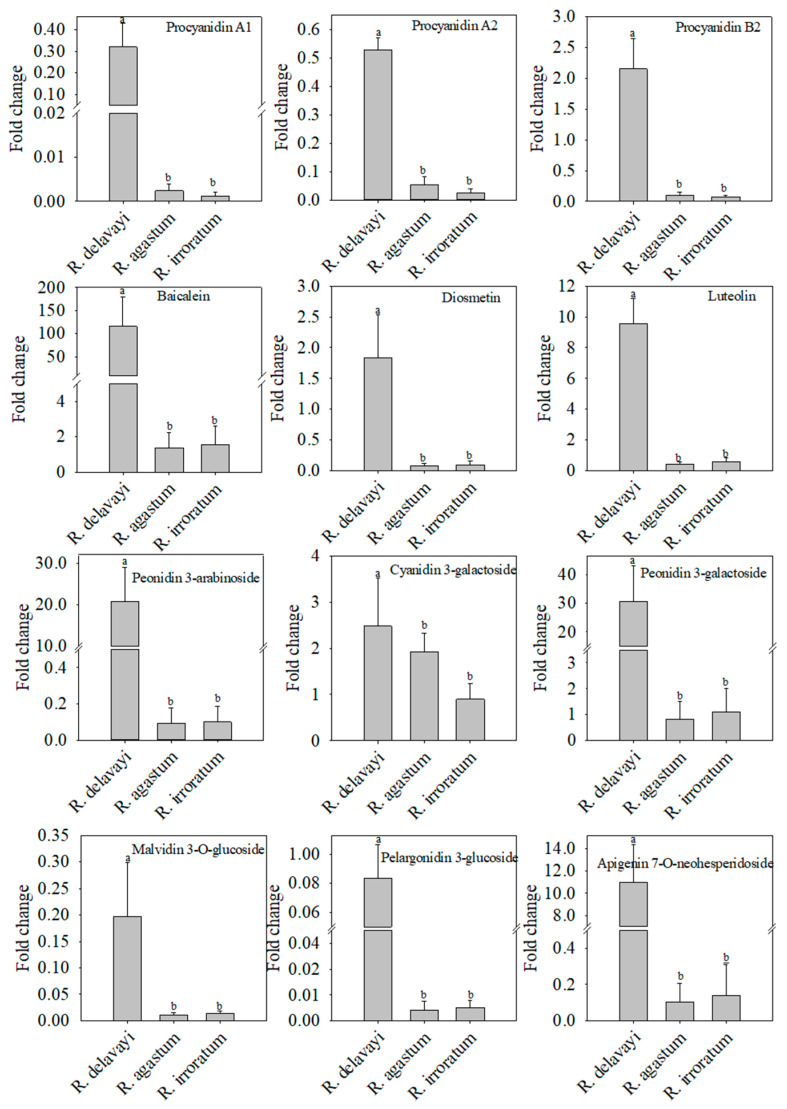
The relative content of the differential flavonoid metabolites in *R. delavayi*, *R. agastum*, and *R. irroratum*. All the metabolites were detected using UHPLC-QTOF-MS in the positive ion mode. The differentially accumulated metabolites among three species were selected based on a Student’s *t*-test with a *p*-value < 0.05 and a Variable Importance in the Projection (VIP) score > 1. The presence of different lowercase letters on the bars indicates statistically significant differences (*p* < 0.05).

**Figure 7 ijms-25-09605-f007:**
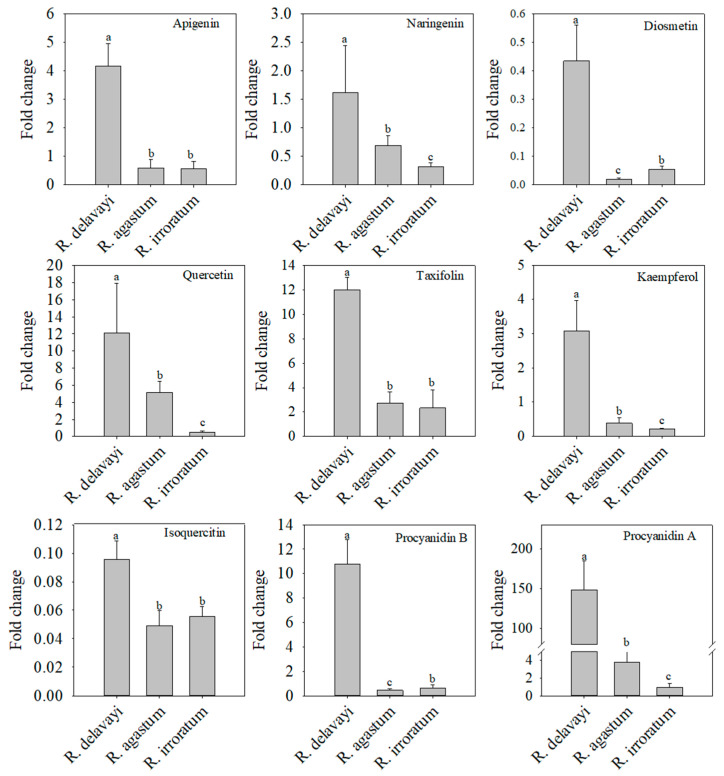
The relative content of the different flavonoid metabolites in *R. delavayi*, *R. agastum*, and *R. irroratum*. All the metabolites were detected using UHPLC-QTOF-MS in the negative ion mode. The differentially accumulated metabolites among three species were selected based on a Student’s *t*-test with a *p*-value < 0.05 and a Variable Importance in the Projection (VIP) score > 1. The presence of different lowercase letters on the bars indicates statistically significant differences (*p* < 0.05).

**Figure 8 ijms-25-09605-f008:**
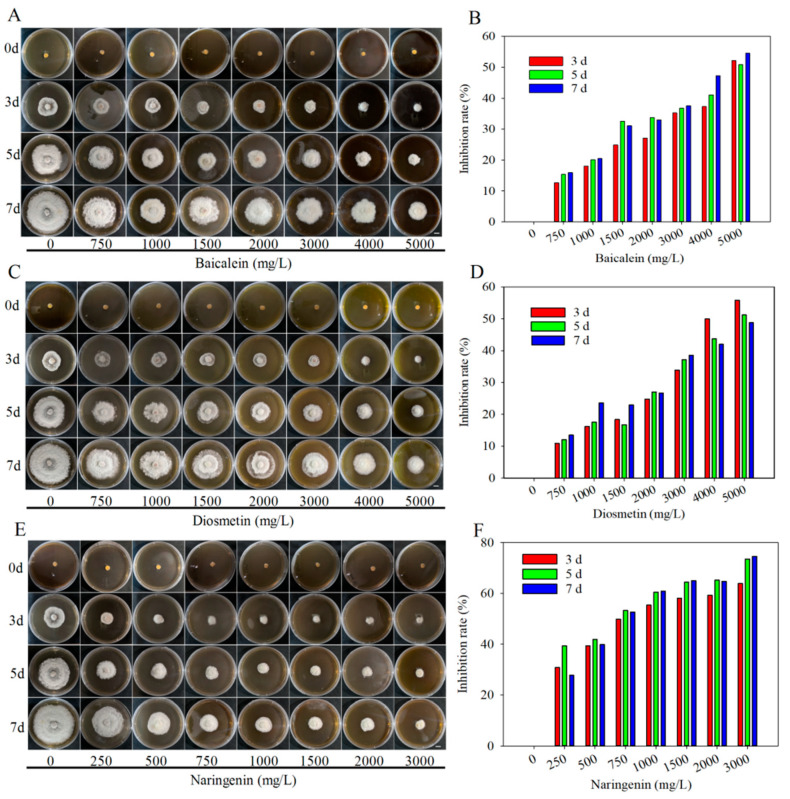
The impact of baicalein, diosmetin, and naringin on the mycelial growth of *N. clavispora*. (**A**,**C**,**E**) depicts the phenotypic effects of varying concentrations of baicalein, diosmetin, and naringin on the mycelial growth of *N. clavispora*, respectively. Bar = 1 cm. (**B**,**D**,**F**) shows the inhibitory rates of the mycelial growth of *N. clavispora* at different concentrations of baicalein, diosmetin, and naringin, respectively.

**Figure 9 ijms-25-09605-f009:**
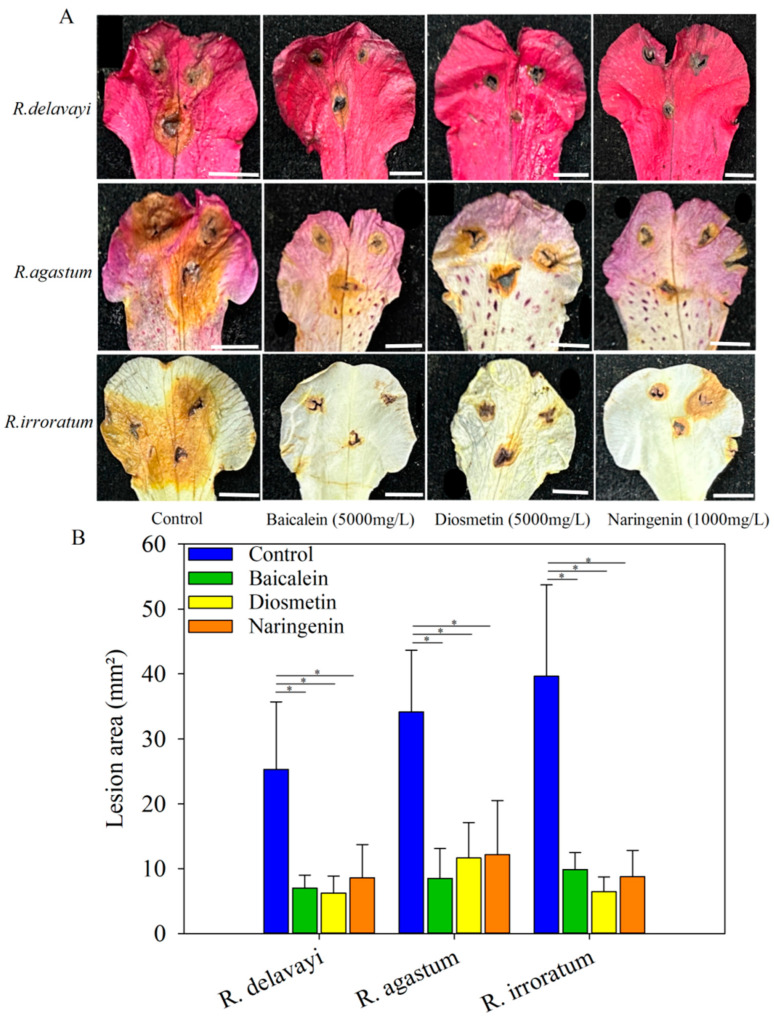
The inhibitory effects of baicalein, diosmetin, and naringin on petal blight disease in *R. delavayi*, *R. agastum*, and *R. irroratum*. (**A**) shows the phenotypic changes in petal blight disease in the three species after the treatments with baicalein, diosmetin, and naringin. Bar = 5 mm. (**B**) displays the lesion areas on the petals of three species following the treatments with control solution (0.01 mol/L NaOH), baicalein (5000 mg/L), diosmetin (5000 mg/L), and naringin (1000 mg/L), respectively. Significance was analyzed using one-way analysis of variance (ANOVA) between the control and the metabolites, and * indicates a significant difference in the lesion area between the treatment with the metabolites and the control (*p* < 0.05).

**Table 1 ijms-25-09605-t001:** The differentially accumulated flavonoids and their antifungal activity according to the published literature.

Metabolites	Chemical Formula	Antifungal Activity	References
Apigenin	C_15_H_10_O_5_	*Alternaria tenuissima*	[21]
Apigenin 7-O-neohesperidoside	C_27_H_30_O_14_	*/*	/
Baicalein	C_21_H_18_O_11_	*Candida albicans*, *Candida tropicalis* and *Candida parapsilosis*	[22]
Cyanidin 3-galactoside	C_21_H_21_O_11_	*/*	/
Diosmetin	C_16_H_12_O_6_	*Bacillus subtilis* and *Trichophyton rubrum*	[23]
Isoquercitin	C_21_H_20_O_12_	*/*	/
Kaempferol	C_15_H_10_O_6_	*C. parapsilosis*	[24]
Luteolin	C_15_H_10_O_6_	*/*	/
Malvidin 3-O-glucoside	C_23_H_25_O_12_	*/*	/
Naringenin	C_15_H_12_O_5_	*Phytophthora nicotianae*	[25]
Naringenin-7-O-Glucoside	C_21_H_22_O_10_	*/*	/
Pelargonidin 3-glucoside	C_27_H_31_O_15_	*/*	/
Peonidin 3-arabinoside	C_21_H_21_O_10_	*/*	/
Peonidin 3-galactoside	C_22_H_23_O_11_	*/*	/
Procyanidin A	C_30_H_24_O_12_	*/*	/
Procyanidin A1	C_30_H_24_O_12_	*/*	/
Procyanidin A2	C_30_H_24_O_12_	*/*	/
Procyanidin B	C_30_H_26_O_12_	*/*	/
Procyanidin B2	C_30_H_26_O_12_	*/*	/
Quercetin	C_15_H_10_O_7_	*C. parapsilosis* and *Colletotrichum gloeosporioides*	[24,26]
Taxifolin	C_15_H_12_O_7_	*Mycosphaerella graminicola*	[27]

## Data Availability

The data presented in this study are available on request from the corresponding author.

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
