# Peer review of "The Preliminary Analysis of Flavonoids in the Petals of Rhododendron delavayi, Rhododendron agastum and Rhododendron irroratum Infected with Neopestalotiopsis clavispora"

_ijms, 2024, doi:10.3390/ijms25179605_

Round 1

Reviewer 1 Report

Comments and Suggestions for Authors

The authors described the role of secondary metabolites in the petals of Rhododendron species against N. clavispora. It is interesting topic and provides some scientific aspects. However, it has been some weak points to publish with current form of manuscript.  

I have pointed out some weak points as follows:

1.    Fig 1-Fig 9, please write full sentence rather than incomplete sentence and add more information based on conducting experiments.  

2.    Fig 2, please enlarge size of main figures.  

3.    Fig 3 and Fig 4, I cannot read anything. Increase figure resolution.

4.    Please double check the space between final word and reference number throughout whole manuscript.

5.    Line68, author mentioned pathogenic fungi but all examples that you listed were bacterial. Please check this out.

6.    Based on result from Fig 2, please explain why disease incidence and index should be increased in R. delavayi under experimental condition compared to that of nature conditions.

7.    Based on your data, I assume that delavayi is a resistant species, agastum is a susceptible, and irroratum should be an intermediate. In Fig 9, lesion area pattern in control treated plants are not the same tendency as compared to Fig 2. Please explain this discrepancy.

8.    Add potential mechanism of inhibitory mechanisms by secondary metabolites that you discovered in discussion part.   

Comments on the Quality of English Language

no comments 

Reviewer 2 Report

Comments and Suggestions for Authors

Dear authors, find attached file.

Round 2

Reviewer 1 Report

Comments and Suggestions for Authors

Revised form should be fine for acceptance.